# An Empirical Analysis of Compute-Optimal Inference for Problem-Solving with Language Models

## Abstract

The optimal training configurations of large language models (LLMs) with respect to model sizes and compute budgets have been extensively studied. But how to optimally configure LLMs during inference has not been explored in sufficient depth. We study *compute-optimal inference*: designing models and inference strategies that optimally trade off additional inference-time compute for improved performance. As a first step towards understanding and designing compute-optimal inference methods, we assessed the effectiveness and computational efficiency of multiple inference strategies such as Greedy Search, Majority Voting, Best-of-N, Weighted Voting, and their variants on two different Tree Search algorithms, involving different model sizes (e.g., 7B and 34B) and computational budgets. We found that a smaller language model with a novel tree search algorithm typically achieves a Pareto-optimal trade-off. These results highlight the potential benefits of deploying smaller models equipped with more sophisticated decoding algorithms in end-devices to enhance problem-solving accuracy. For instance, we show that the Llemma-7B model can achieve competitive accuracy to a Llemma-34B model on MATH500 while using $2\times$ less FLOPs. Our findings could potentially apply to any generation task with a well-defined measure of success.

## 1 Introduction

Scaling laws of neural networks [Hestness et al., 2017, Rosenfeld et al., 2019] have been established across a range of domains, including language modeling [Kaplan et al., 2020, Hoffmann et al., 2022, OpenAI, 2023], image modeling [Henighan et al., 2020, Yu et al., 2022, Peebles and Xie, 2023], video modeling [Brooks et al., 2024], reward modeling [Gao et al., 2023], and board games [Jones, 2021]. These studies have demonstrated how model performance is influenced by both the size of the model and the amount of training computation. However, there is limited knowledge on how varying the compute during inference affects model performance after the model has been trained.

To improve the task performance of large language models (LLMs), inference techniques typically involve additional computation in a *performance maximization* step at inference time [Nye et al., 2021, Wei et al., 2022, Wang et al., 2022b, Yao et al., 2023, Chen et al., 2024b]. This cost must be taken into account for *compute-optimal inference.* For example, a Monte Carlo Tree Search (MCTS) method [Jones, 2021] may improve task performance, but potentially cost much more than simply sampling solutions multiple times. Generally speaking, we need a comprehensive understanding of how various inference-time methods (e.g., Best-of-N, majority voting) trade off between performance and cost. To improve our understanding, this paper presents a thorough empirical evaluation with careful analysis over various configurations of representative LLMs and inference algorithms.

Specifically, we explore how to select an optimal model size (e.g., 7B or 34B) for the policy model and an effective inference strategy (e.g., Greedy Search, Majority Voting, Best-of-N, Weighted Voting,

Submitted to 38th Conference on Neural Information Processing Systems (NeurIPS 2024). Do not distribute.

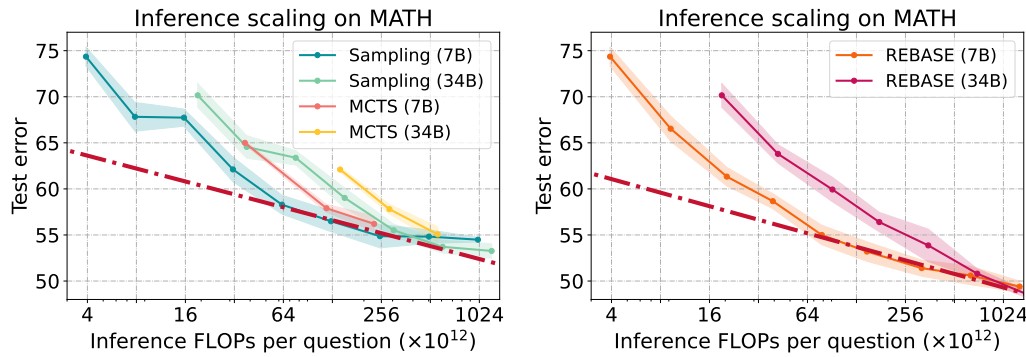

Figure 1: **The inference computation scaling laws** exhibited in error rate on the **MATH500** test set based on weighted majority voting, where the left figure shows sampling vs. MCTS, and the right figure shows our proposed REBASE. Clearly, the error rate decreases steadily when the computation increases, and REBASE exhibits a Pareto-optimal tradeoff during inference.

and their Tree Search variants) to maximize performance (i.e., accuracy) within a given compute budget. We manipulate the inference computation (FLOPs) of a fixed model by generating additional tokens through the policy model, sampling further candidate solutions, and ranking them with a reward model. We analyze the performance of a family of math-specialized LLMs (i.e., Llemma-7B and Llemma-34B [Azerbayev et al., 2023]) fine-tuned on the MetaMath dataset [Yu et al., 2023] and measure the error rate on the GSM8K test set [Cobbe et al., 2021a] and MATH500 test set [Hendrycks et al., 2021b, Lightman et al., 2023b].

Our analysis shows that voting-based methods generally outperform the strategy which selects the best solution (i.e., Best-of-N), and weighted voting has the most favorable results (Section 4.3, Figure 5 & 6). However, neither method shows a desirable behavior at high levels of compute. For instance, weighted voting saturates when sampling more than 128 solutions (Figure 1). We have also found that the commonly used MCTS method does not perform well with weighted voting, as it often yields many unfinished solutions, hence having less votes. To address this issue, we propose a novel tree search algorithm, *REward BAlanced SEarch (*REBASE*)*, which pairs well with weighted voting and improves the Pareto-optimal trade-off between accuracy and inference compute. The key idea of REBASE is to use a node-quality based reward to control the exploitation and pruning properties of tree search, while ensuring enough candidate solutions for voting or selection.

In our experiments, REBASE consistently outperforms sampling and MCTS methods across all settings, models, and tasks. Importantly, we find that REBASE with a *smaller* language model typically achieves a Pareto-optimal trade-off. For instance, we show that the Llemma-7B model can achieve competitive accuracy to a Llemma-34B model while using $2\times$ less FLOPs when evaluating on MATH500 (Figure 1) or GSM8K (Figure 4). These findings underscore the advantages of using smaller models with advanced inference-time algorithms on end-devices to improve problem-solving.

## 2 Related Works

**Mathematical Reasoning with LLMs.** Large language models have made significant progress in recent years, and have exhibited strong reasoning abilities [Brown et al., 2020, Hoffmann et al., 2022, Chowdhery et al., 2022, Lewkowycz et al., 2022]. Mathematical problem solving is a key task for measuring LLM reasoning abilities [Cobbe et al., 2021a, Hendrycks et al., 2021b]. [Ling et al., 2017] first developed the method of producing step by step solutions that lead to the final answer. Later, [Cobbe et al., 2021b] extended the work by finetuning the pre-trained language model on a large dataset to solve math word problems, a verifier is trained for evaluating solutions and ranking solutions. Nye et al. [2021] train models to use a scratchpad and improve their performance on algorithmic tasks. Wei et al. [2022] demonstrate that the reasoning ability of a language model can be elicited through the prompting. Subsequent research [Kojima et al., 2022, Lewkowycz et al., 2022, Zhou et al., 2022] in reasoning tasks has also highlighted the efficacy of rationale augmentation. We choose problem solving in mathematics as the task to study the compute-optimal strategy since it allows us to accurately evaluate the problem solving ability of LLMs.

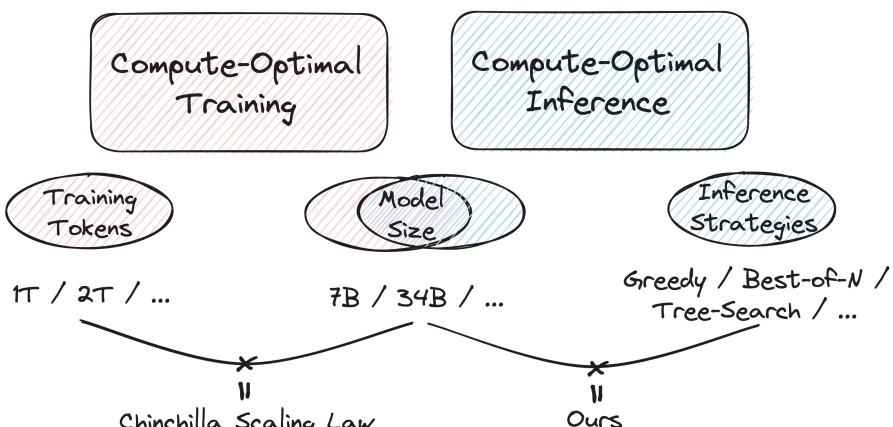

Figure 2: **Illustration of compute-optimal scaling laws in training and inference.** The Chinchilla scaling law shows how to choose a model size and number of training tokens under a training-compute budget, while ours shows how to choose a model size and an inference strategy under a inference-compute budget.

**Inference Strategies of LLM Problem Solving.** A variety of inference (also called decoding) strategies have been developed to generate sequences with a trained model. Deterministic methods such as greedy decoding and beam search [Teller, 2000, Graves, 2012] find highly probable sequences, often yielding high quality results but without diversity. Sampling algorithms (e.g., temperature sampling [Ackley et al., 1985]) can produce a diverse set of results which are then aggregated to achieve higher accuracy (e.g., via majority voting [Wang et al., 2022a]). Recent methods combine search algorithms with modern LLMs, including breadth-first or depth-first search [Yao et al., 2023], Monte-Carlo Tree Search (MCTS) [Zhang et al., 2023, Zhou et al., 2023, Liu et al., 2024, Choi et al., 2023], and Self-evaluation Guided Beam Search [Xie et al., 2023]. All of these methods show that using search at inference time can lead to performance gains in various tasks. However, the trade-off for the improved performance is the use of compute to perform the search. Analyzing the trade-off between compute budget and LLM inference performance remains understudied. In this paper, we systematically analyze the trade-off between compute budget and problem-solving performance, and propose a tree search method that is empirically Pareto-optimal.

**Process Reward Models.** Process reward models (PRMs) have emerged as a technique to improve the reasoning and problem-solving capabilities of LLMs. These models assign rewards to the intermediate steps of the LLM generated sequences. PRMs have been shown effective in selecting reasoning traces with a low error rate, and for providing rewards in reinforcement learning-style algorithms [Uesato et al., 2022, Polu and Sutskever, 2020, Gudibande et al., 2023]. Ma et al. [2023] applies the PRM to give rewards on the intermediate steps and guide the multi-step reasoning process. The PRM can be either trained on human labeled data [Lightman et al., 2023a] or model-labeled synthetic data [Wang et al., 2023]. In our work, we use the PRM as the reward model for selecting generated solutions, and for selecting which partial solutions to explore in tree search.

## 3 An Empirical Analysis of Compute-Optimal Inference for Problem-Solving

We explore the following question: *Given a fixed FLOPs budget, how should one select an optimal model size for the policy model, and an effective inference strategy to maximize performance (i.e., accuracy)?* To address this, we represent the problem-solving error rate $E(N, T)$ as a function of the number of model parameters $N$ and the number of generated tokens $T$. The computational budget $C$ is a deterministic function $\text{FLOPs}(N, T)$, based on $N$ and $T$. Our goal is to minimize $E$ under the test-time compute constraint $\text{FLOPs}(N, T) = C$:

$$N_{opt}(C), T_{opt}(C) = \underset{N, T \text{ s.t. FLOPs}(N,T)=C}{\arg\min} E(N, T) \tag{1}$$

where $N_{opt}(C)$ and $T_{opt}(C)$ denote the optimal allocation of a computational budget $C$.

Here, the inference computation (FLOPs) for a fixed model can be modulated by generating more tokens with the policy model, e.g., by sampling additional candidate solutions and subsequently ranking them using a reward model. We primarily consider sampling and tree-search approaches with reranking or majority voting as the means to consume more tokens, including Greedy Search, Majority Voting, Best-of-N, Weighted Voting, and their variants on tree search methods.

### 3.1 Inference Strategies

#### 3.1.1 Sampling

**Greedy Search.** This strategy generates tokens one at a time by selecting the highest probability token at each step, without considering future steps. It is computationally efficient but often suboptimal in terms of diversity.

**Best-of-n.** This strategy, also known as rejection sampling, samples multiple solutions and chooses the one with the highest score given by the reward model.

**Majority Voting.** In this strategy, multiple model outputs are generated, and the final answer to the problem is determined by the most frequently occurring answer in all the outputs.

**Weighted Majority Voting.** This strategy is a variant of majority voting in which the votes are weighted based on the score given by the reward model.

#### 3.1.2 Monte Carlo Tree Search (MCTS)

Monte Carlo Tree Search (MCTS) has proven effective in domains such as board games where strategic decision-making is required [Silver et al., 2016, 2017, Jones, 2021]. Recent work has shown that adapting MCTS to the context of LLMs can enhance the text generation process [Zhang et al., 2023, Zhou et al., 2023, Liu et al., 2024, Choi et al., 2023, Chen et al., 2024a, Tian et al., 2024, Chen et al., 2024a]. In this context, MCTS is often paired with a value model to score and guide the exploration steps. For additional background, we provide a review of MCTS in Appendix B.

Recent work in MCTS or its variants (e.g., Tree of Thoughts [Yao et al., 2023]) mainly focus on improving the performance (e.g., accuracy) on the studied tasks. However, generic comparisons of MCTS with conventional methods like Best-of-n and Majority Voting in terms of computational budget, measured in generated tokens or processing time, are either scarce or indicating inference-time issues. For example, MCTS consumes substantially more resources, often requiring dozens of times more generated tokens than simpler methods. Specifically, a significant portion of the paths in the search tree are used to estimate and select nodes, and these paths do not necessarily become a part of the final candidate solution, although MCTS ensures that the sampled solutions comprise high-quality intermediate steps. In contrast, sampling methods generate multiple solutions in parallel and independently, and all the generated sequences are included in the candidate solutions. However, the intermediate steps in these sequences are not guaranteed to be of high quality, as there is no mechanism for pruning poor steps or exploiting promising ones.

This highlights the need for developing a new tree search method that can achieve a comparable (or better) performance as MCTS, and that is computationally less costly, just like weighted majority voting and best-of-n. This need leads to the development of our new method named Reward Balanced SEarch (REBASE), as introduced next.

#### 3.1.3 Reward Balanced Search (REBASE)

The REBASE tree search method inherits the exploitation and pruning properties of tree search, while using the reward model alone to estimate the nodes' qualities without additional computation for estimating values by sampling children. The efficiency is achieved by constraining the total expansion width of the tree at a certain depth. REBASE balances the expansion width among the nodes at the same depth based on the rewards given by the Process Reward Model (PRM). The details are provided below:

**Notations.** We consider the fine-tuned LLM as a policy $\pi_\theta$. Given a question $q$ and the first $k$ steps of a solution $x_1, \cdots, x_k$, the $(k+1)$-th step is produced by $\pi_\theta(x_{k+1}|q, x_1 \cdots x_k)$. When generating solutions using tree search, the root of the tree corresponds to the question $q$. The node corresponding

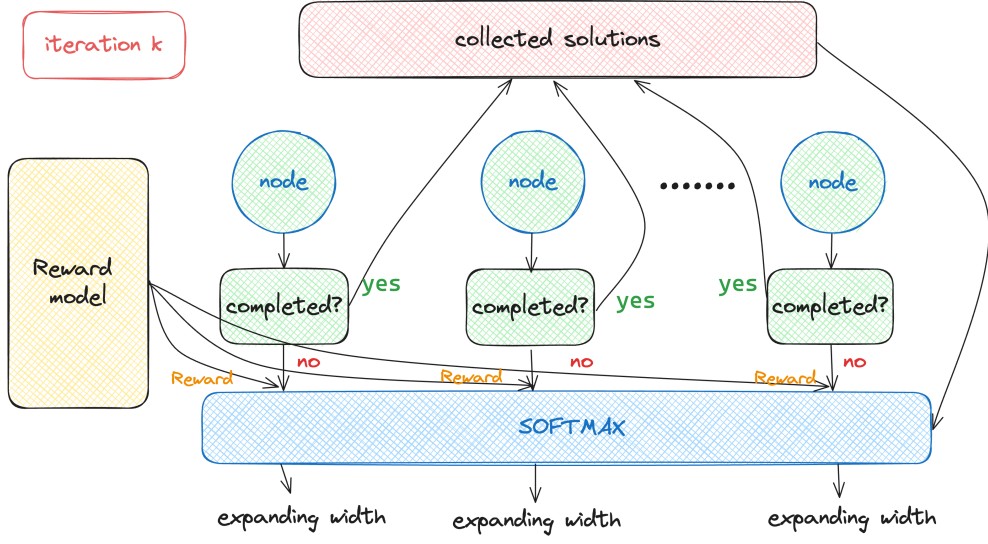

Figure 3: Illustration of one iteration of REward BAlanced SEarch (REBASE).

to $x_{k+1}$ is the child of the node corresponding to $x_k$ if it is sampled from $\pi_\theta(\cdot|q, x_1 \cdots, x_k)$. The reward of a node $n(x_k)$ is determined by the PRM as $R(n(x_k)) = R(q, x_1, \cdots, x_k)$.

**Initialization.** Given the question $q$, balance temperature $T_b$, and sampling number of solutions N, we sample N instances of the first step for the question, yielding all the nodes of depth 1 in the search tree. We set the sampling budget of depth 0 $B_0 = N$ as initialization.

**Reward modeling and update.** In the $i$-th iteration, the PRM assigns the rewards to all the nodes at depth $i$. After that, the algorithm examines whether the solutions up to depth $i$ are complete. Supposing there are $C_i$ completed solutions, we update the sampling budget using $B_i \leftarrow B_{i-1} - C_i$. If $B_i = 0$, the process ends, and we obtain $N$ solutions.

**Exploration balancing and expansion.** For all of the nodes $n_j$ with reward $R(n_j)$ in the depth $i$ of the tree, we calculate the expansion width of the $n_j$ as:

$$W_j = \text{Round}\left(B_i \frac{\exp\left(R(n_j)/T_b\right)}{\sum_k \exp\left(R(n_k)/T_b\right)}\right). \tag{2}$$

Then we sample $W_j$ children for $n_j$ for all the nodes in depth $i$, and start the next iteration.

### 3.1.4 Theoretical Analysis

Before empirically studying the scaling effects of increasing the inference-time compute budget, we present two theorems which will help us understand the experimental results later. These two theorems give an upper bound on the performance of sampling when fixing the LLM generator.

We assume the vocabulary is limited and the sequence length is constrained, thus the number of possible solutions and answers are finite. The proofs are provided in the Appendix A.

**Theorem 1.** *Given a test dataset $\mathcal{D}$ and a LLM $\pi$. $|\mathcal{A}|$ is the finite set of all possible answers given by LLM, the ground truth function $g$ maps test data $d$ to the true answer. Denote the accuracy of the LLM on this dataset with majority over N samples as $ACC_{MV}(\pi, \mathcal{D}, N)$. The accuracy of majority voting on the LLM will eventually saturate, i.e.*

$$\lim_{N \to \infty} ACC_{MV}(\pi, \mathcal{D}, N) = \frac{\sum_{d \in \mathcal{D}} \mathbb{I}\left((g(d) = \arg\max_{a \in \mathcal{A}} \pi(a|d)\right)}{|\mathcal{D}|}. \tag{3}$$

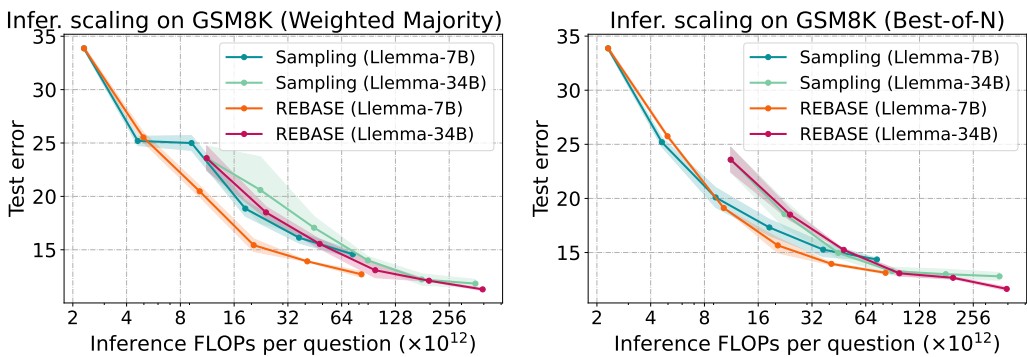

Figure 4: **The inference computation scaling comparisons across different model sizes**. The left/right panel shows the GSM8K problem-solving error rate on GSM8K based on Weighted Mjority/Best-of-N.

where $\pi(x|d)$ denotes the probability that the LLM answers $x$ given input $d$ and $\mathbb{I}$ is the indicator function.

**Theorem 2.** *Assume the reward model assigns an expected reward of $R(a)$ to $a \in \mathcal{A}$ among the different solutions generated by LLM that yields $a$. Given a test dataset $\mathcal{D}$ and a LLM $\pi$. $|\mathcal{A}|$ is the finite set of all possible answers given by LLM, the ground truth function $g$ maps test data $d$ to the true answer. Denote the accuracy of the LLM on this dataset with weighted majority over N samples as $ACC_{WV}(\pi, \mathcal{D}, N, R)$. The accuracy of weighted majority voting on the LLM will eventually saturate, i.e.*

$$\lim_{N \to \infty} ACC_{WV}(\pi, \mathcal{D}, N, R) = \frac{\sum_{d \in \mathcal{D}} \mathbb{I}((g(d) = \arg\max_{a \in \mathcal{A}} R(a)\pi(a|d))}{|\mathcal{D}|}. \tag{4}$$

*where $\pi(x|d)$ denotes the probability that the LLM answers $x$ given input $d$ and $\mathbb{I}$ denotes the indicator function.*

Theorem 2 shows that as long as the reward model assigns higher rewards than the policy for correct answers versus other answers in expectation, the upper bound of Weighted Majority Voting will be higher than Majority Voting since $\mathbb{I}((g(d) = \arg\max_{a \in \mathcal{A}} R(a)\pi(a|d)) > \mathbb{I}((g(d) = \arg\max_{a \in \mathcal{A}} \pi(a|d))$. We put the figures comparing BoN and Weighted Majority Voting in the main paper and leave the Majority Voting figures in Appendix D since Majority Voting is dominated by Weighted Majority Voting.

## 4 Experiments

### 4.1 Setup

**Datasets.** We conduct experiments on two mathematical problem-solving datasets to investigate the scaling effects of computation and our REBASE method for both challenging and simpler problems. Specifically, MATH [Hendrycks et al., 2021a] and GSM8K[Cobbe et al., 2021b] are datasets containing high school mathematics competition-level problems and grade-school level mathematical reasoning problems, respectively. Following [Lightman et al., 2023b, Wang et al., 2024, Sun et al., 2024], we use the MATH500 subset as our test set.

**Generators.** We use Llemma-7B and Llemma-34B [Azerbayev et al., 2024] as our base models and finetune them on the MetaMath dataset [Yu et al., 2024] using full parameter supervised fine-tuning (Full-SFT), The detailed finetuning configuration is given in the Appendix. Additionally, we test the Mistral-7B model to expand our findings across different models.

**Reward Model.** All of the experiments use the same Llemma-34B reward model, which we finetuned on the synthetic process reward modeling dataset, Math-Shepherd [Wang et al., 2024]. We added a reward head to make the model, enabling it to output a scalar reward at the end of each step.

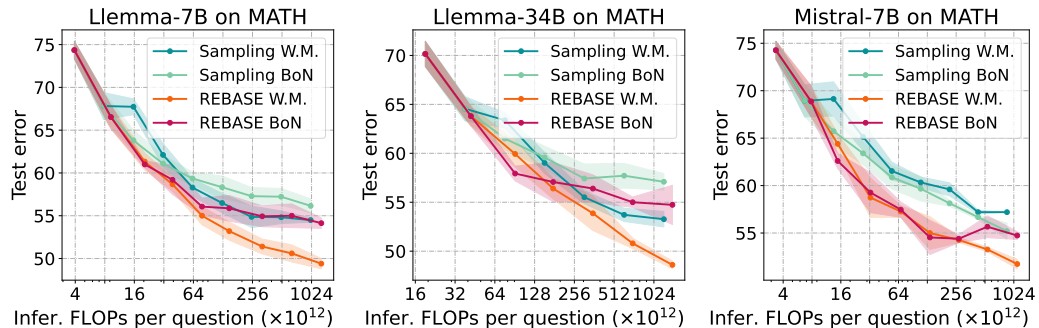

Figure 5: **The inference computation scaling laws** of different models for the problem-solving error rate on **MATH500** test set. The tested models are Llemma-7B (left), Llemma-34B (middle), & Mistral-7B (right). In the legend, W.M. and BoN refer to Weighted Majority and Best-of-N, respectively.

**Inference Configuration.** For the MATH dataset, we sample 1, 2, 4, 8, 16, 32, 64, 128, and 256 solutions for the 7B models, and 1 to 64 solutions for the 34B Llemma model. For the GSM8K dataset, we sample 1 to 32 solutions, as it is relatively easier. We use sampling and REBASE to generate these samples and select the answer through Best-of-N, Majority Voting, and Weighted Voting. Each configuration is run multiple times to calculate the mean and variance, thereby mitigating the randomness and ensuring the reliability of our conclusions.

## 4.2 Main Results of Compute-Optimal Inference

In order to compare the compute budgets of 7B and 34B models, we plot the figures with the number of FLOPs used per question during inference. We compute the inference FLOPs based on the standard formula from [Kaplan et al., 2020].

**Llemma-7B model achieves competitive accuracy to Llemma-34B model with lower compute budget.** Figures 1 and 4 show the curves of error rates versus total number of inference FLOPs per question. Inference methods with different model sizes are plotted in the same diagram. We found that Llemma-7B costs approximately 2x less total FLOPs than Llemma-34B under the same method (Sampling, MCTS, REBASE) and task (MATH, GSM8K) while achieving competitive accuracy. This result suggests that, with the same training dataset and model family, training and inference with a smaller model could be more favorable in terms of compute budget if multiple sampling or search methods are employed.

**All inference configurations will saturate eventually.** This is expected as Theorem 1 and Theorem 2 show. Also illustrated in Figures 5 and 6, the slope of the erro rate curves start large, then decreases and the curves finally become nearly flat as the number of samples scales, showing the effect of saturation.

**Scaling law of compute-optimal inference.** The findings in our experiments are consistent with the Theorem 1 and 2, After a threshold the accruacy of sampling more solutions saturate, we should scale the model size. We interpolate the smoothed test error rate curve in Figure 1 and Figure 4, and fit power laws to estimate the optimal model size $N$ and number of generated tokens $T$ for any given amount of compute. We obtained a relationship $N_{opt} \propto C^a$ and $T_{opt} \propto C^b$, where $a = 1.0$ and $b = 0.0$ for both sampling-based weighted voting and our tree-search method REBASE. Our fitted curves indicate that the optimal inference strategy is invariant to the amount of compute (e.g., re-ranking with 32 sampled solutions or REBASE tree search with a compute budget of 64 for MATH), and the optimal model size grows linearly with the increased compute budget.

## 4.3 Comparing REBASE to Other Baselines

**REBASE is Pareto-optimal.** While MCTS undeperforms Sampling (Fig. 1), from Fig. 1, 4, 5, and 6, we found that REBASE consistently outperforms the Sampling method in all settings, when

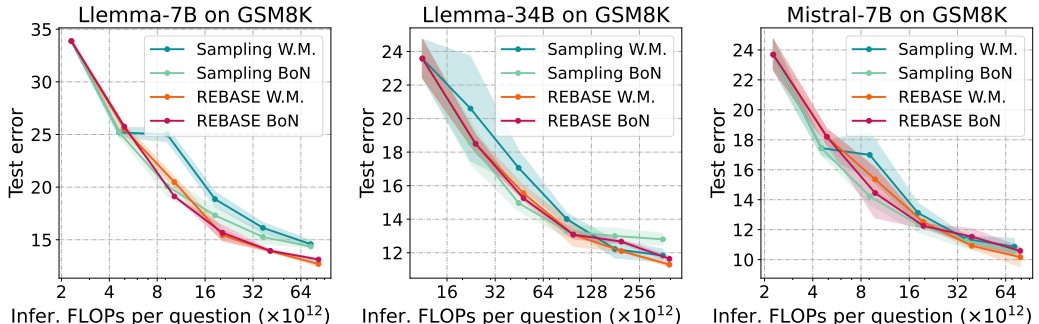

Figure 6: **The inference computation scaling laws** of different models for the problem-solving error rate on **GSM8K** test set. The tested models are Llemma-7B (left), Llemma-34B (middle), & Mistral-7B (right). In the legend, W.M. and BoN refer to Weighted Majority and Best-of-N, respectively.

Table 1: REBASE with lower compute budget has competitive accuracy against Sampling with higher compute budget. We use weighted voting to aggreagte the candidate solutions in both Sampling and REBASE.

|  | # SAMPLES | FLOPS | MATH500 |
|---|---|---|---|
| MISTRAL-7B | | | |
| SAMPLING | 256 | $8.7 \times 10^{14}$ | 42.8 |
| REBASE | 32 | $\mathbf{1.36 \times 10^{14}}$ | **45.0** |
| LLEMMA-7B | | | |
| SAMPLING | 256 | $10.0 \times 10^{14}$ | 45.5 |
| REBASE | 32 | $\mathbf{1.48 \times 10^{14}}$ | **46.8** |
| LLEMMA-34B | | | |
| SAMPLING | 64 | $12.1 \times 10^{14}$ | 46.7 |
| REBASE | 32 | $\mathbf{7.08 \times 10^{14}}$ | **49.2** |

fixing the model and the evaluation task. Table 1 shows that REBASE can achieve competitive accuracy with even a lower compute budget than the sampling method. This finding is novel, and differs from previous tree search works which typically improve the performance at the cost of higher computational expense compared to sampling [Chen et al., 2024a, Xie et al., 2023]. Table 2 shows that given the same compute budget (sampling 32 solutions for the 7B model and 8 solutions for 34B model), using REBASE yields higher accuray than sampling.

**Weaker models gain more from Tree Search.** From Fig. 2, we saw that compared with sampling, Mistral-7B, Llemma-7B, Llemma-34B increase 5.3%, 3.3%, 2.6% in MATH and 0.7%, 1.9%, 0.9% in GSM8K. The order of accuracy increase is inversely related to the model's corresponding greedy search on those datasets. This suggests that weaker models, as indicated by their lower greedy search accuracy, benefit more from tree search methods like REBASE.

**REBASE saturates later than sampling with higher accuray.** From Figure 5 and Figure 6, we observe that both sampling and REBASE saturate early in GSM8K and relatively late in MATH, which we attribute to the difference of the difficulty level. This can be explained through the LLM may assign high probability to the true answer in easy problems than those of harder problem, as suggested by Theorem 1 and 2 with their proofs A. On MATH (Figure 5), we see that REBASE finally saturates with a higher accuracy than sampling. We hypothesize the reason is that REBASE samples the truth answer with higher probability than sampling. And as demonstrated by Theorem 1 and 2, the upper bound becomes higher.

Table 2: Accuracy of diffrent inference configurations under a specific compute budget. MV, BoN and WV denote Majority Voting, Best-of-N and Weighted Voting, respectively.

| | # SAMPLES | MATH FLOPs | GSM8K FLOPs | MATH500 | GSM8K |
|---|---|---|---|---|---|
| MISTRAL-7B | | | | | |
| GREEDY | 1 | $3.4 \times 10^{12}$ | $2.3 \times 10^{12}$ | 28.6 | 77.9 |
| SAMPLING + MV | 32 | $109.2 \times 10^{12}$ | $72.6 \times 10^{12}$ | 36.1 | 85.7 |
| SAMPLING + BoN | 32 | $109.2 \times 10^{12}$ | $72.6 \times 10^{12}$ | 40.3 | 89.4 |
| SAMPLING + WV | 32 | $109.2 \times 10^{12}$ | $72.6 \times 10^{12}$ | 39.7 | 89.1 |
| REBASE + MV | 32 | $136.2 \times 10^{12}$ | $78.9 \times 10^{12}$ | 44.1 | 88.8 |
| REBASE + BoN | 32 | $136.2 \times 10^{12}$ | $78.9 \times 10^{12}$ | **45.4** | 89.4 |
| REBASE + WV | 32 | $136.2 \times 10^{12}$ | $78.9 \times 10^{12}$ | 45.0 | **89.8** |
| LLEMMA-7B | | | | | |
| GREEDY | 1 | $3.92 \times 10^{12}$ | $2.3 \times 10^{12}$ | 30.0 | 68.5 |
| SAMPLING + MV | 32 | $125.4 \times 10^{12}$ | $73.9 \times 10^{12}$ | 41.0 | 80.0 |
| SAMPLING + BoN | 32 | $125.4 \times 10^{12}$ | $73.9 \times 10^{12}$ | 41.7 | 85.6 |
| SAMPLING + WV | 32 | $125.4 \times 10^{12}$ | $73.9 \times 10^{12}$ | 43.5 | 85.4 |
| REBASE + MV | 32 | $148.0 \times 10^{12}$ | $82.6 \times 10^{12}$ | 46.1 | 86.1 |
| REBASE + BoN | 32 | $148.0 \times 10^{12}$ | $82.6 \times 10^{12}$ | 44.1 | 86.9 |
| REBASE + WV | 32 | $148.0 \times 10^{12}$ | $82.6 \times 10^{12}$ | **46.8** | **87.3** |
| LLEMMA-34B | | | | | |
| GREEDY | 1 | $19.0 \times 10^{12}$ | $11.2 \times 10^{12}$ | 33.0 | 78.4 |
| SAMPLING + MV | 8 | $152.3 \times 10^{12}$ | $89.7 \times 10^{12}$ | 39.9 | 84.3 |
| SAMPLING + BoN | 8 | $152.3 \times 10^{12}$ | $89.7 \times 10^{12}$ | 40.4 | 86.7 |
| SAMPLING + WV | 8 | $152.3 \times 10^{12}$ | $89.7 \times 10^{12}$ | 41.0 | 86.0 |
| REBASE + MV | 8 | $176.8 \times 10^{12}$ | $98.7 \times 10^{12}$ | **43.9** | 86.1 |
| REBASE + BoN | 8 | $176.8 \times 10^{12}$ | $98.7 \times 10^{12}$ | 43.6 | **86.9** |
| REBASE + WV | 8 | $176.8 \times 10^{12}$ | $98.7 \times 10^{12}$ | 42.9 | **86.9** |

## 5 Conclusion & Limitations

In this work, we have conducted a comprehensive empirical analysis of compute-optimal inference for problem-solving with language models. Our study has revealed several key findings. First, with an optimal inference configuration, a small language model can achieve competitive accuracy to a $4\times$ larger model while using approximately $2\times$ less total FLOPs under the same inference method (Sampling, MCTS, REBASE) and task (MATH, GSM8K), suggesting that training and inference with smaller models could be more favorable in terms of compute budget when combined with multiple sampling or search strategies. Second, our new REBASE tree-search method consistently outperforms sampling (and MCTS) across all settings, models, and tasks, achieving competitive accuracy with even lower compute budget compared to sampling. Our findings highlight the potential of deploying smaller models equipped with more sophisticated inference strategies like REBASE to enhance problem-solving accuracy while maintaining computational efficiency.

**Limitations** First, our experiments focused specifically on mathematical problem-solving tasks, and the generalizability of our findings to other domains remains to be explored. Second, we only investigated a limited range of model scales, primarily focusing on 7B and 34B models. Future research could extend our analysis to a wider range of model sizes to gain a more comprehensive understanding of the scaling laws for compute-optimal inference. Finally, our experiments mainly utilized the MetaMath dataset for training the models. It would be valuable to explore the impact of different training datasets on the performance and efficiency of compute-optimal inference strategies for mathematical problem-solving.

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

## A  Proofs of Theorem 1 and 2

### A.1  Proof of Theorem 1

*Proof.* Suppose the possible answers of the LLM are $x_1, x_2, x_3, \cdots, x_{|\mathcal{A}|}$, with $\pi(x_1|d) > \pi(x_2|d) > \cdots > \pi(x_{|\mathcal{A}|}|d)$. After sampling $N$ solutions from the LLM, we denote the occurence of $x_i$ as $f(x_i)$, the probability that $x_1$ is not the most frequent output is

$$P(f(x_1) \neq \arg\max_x f(x)) \tag{5}$$

With Union bound, we get

$$P(x_1 \neq \arg\max_x f(x)) \tag{6}$$

$$\leq \sum_{i=2}^{|\mathcal{A}|} P(f(x_1) \leq f(x_i)) \tag{7}$$

$$\leq |\mathcal{A}| P(f(x_1) \leq f(x_2)) \tag{8}$$

$$= |\mathcal{A}| \left(1 - P\left(f(x_1) \geq f(x_2)\right)\right) \tag{9}$$

$$\leq |\mathcal{A}| \left(1 - P\left(f(x_1) \geq \frac{\pi(x_1|d) + \pi(x_2|d)}{2} N\right) P\left(f(x_2) \leq \frac{\pi(x_1|d) + \pi(x_2|d)}{2} N\right)\right) \tag{10}$$

$$\leq |\mathcal{A}| \left(1 - \left(1 - e^{-\frac{\delta_1^2}{2}\pi(x_1|d)N}\right)\left(1 - e^{-\frac{\delta_2^2}{2+\delta_2}\pi(x_2|d)N}\right)\right) \tag{11}$$

$$\leq |\mathcal{A}| C^N \quad \text{for some } C < 1. \tag{12}$$

Where (11) is by Chernoff Bound, $\delta_1 = \frac{\pi(x_1|d) - \pi(x_2|d)}{2\pi(x_1|d)}$ and $\delta_2 = \frac{\pi(x_1|d) - \pi(x_2|d)}{2\pi(x_2|d)}$. As $N \to \infty$, we have

$$f(x) = \begin{cases} M(x|N) = 1 & \text{if } x = \arg\max_{a \in \mathcal{A}} \pi(a|d) \\ M(x|N) = 0 & \text{otherwise}. \end{cases} \tag{13}$$

Where $M(x|N)$ denotes the probability that majority voting over $N$ sampled solutions gives $x$. The proof of original theorem is automatically completed by (13). □

### A.2  Proof of Theorem 2

*Proof.* The proof is similar to the Theorem 1, We rank $x_1, x_2, \cdots, x_{|\mathcal{A}|}$ with $R(x_1)f(x_1) > \cdots > R(x_{|\mathcal{A}|})f(x_{|\mathcal{A}|})$. Denotes $w(x_i)$ as the the total weights of answer $x_i$ after sampling N solutions. As $N \to \infty$, $w(x_i) \to R(x_i)f(x_i)$. Same as proof in theorem 1, we have

$$P(x_1 \neq \arg\max_x f(x)) \tag{14}$$

$$\leq |\mathcal{A}| P(w(x_1) \leq w(x_2)) \tag{15}$$

$$= |\mathcal{A}| \left(1 - P\left(w(x_1) \geq w(x_2)\right)\right) \tag{16}$$

$$\leq |\mathcal{A}| \left(1 - P\left(w(x_1) \geq \frac{v(x_1) + v(x_2)}{2} N\right) P\left(w(x_2) \leq \frac{v(x_1) + v(x_2)}{2} N\right)\right). \tag{17}$$

Where $v(x) = R(x)\pi(x|d)$, the remaining proof completely follows Theorem 1. □

## B  MCTS Details

The MCTS process can be represented as the following steps:

**Selection**  The process begins at the root node. Here, the algorithm recursively selects the child node that offers the highest Upper Confidence Bound applied to Trees (UCT) value, continuing until a node is reached that has not been expanded. The UCT is calculated using the formula

$$UCT(s) = Q(s) + C\sqrt{\frac{\ln\left(N(Parent(s))\right)}{N(s)}}. \tag{18}$$

Table 3: Fine-tuning Hyper-parameters: LR refers to the learning rate, BS refers to the batch size. Llemma-7B and LLemma-34B are the generators we use in our experiments, RM is short for Reward Model.

| Model | # Epoch | Dataset | BS | LR | Max Seq Length | Dtype |
|---|---|---|---|---|---|---|
| Llemma-7B | 1 | MetaMath | 128 | 8E-6 | 1024 | FP32 |
| Llemma-34B | 1 | MetaMath | 128 | 8E-6 | 768 | FP32 |
| Llemma-34B RM | 2 | Math-Shepherd | 128 | 1E-5 | 768 | BF16 |

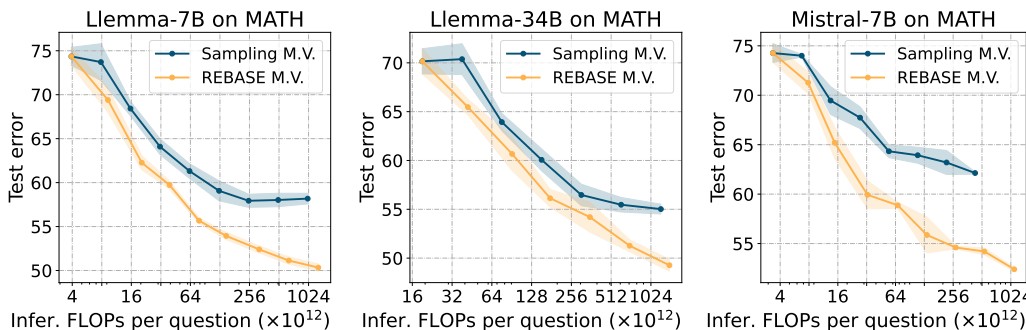

Figure 7: **The inference computation scaling laws** of different models for the problem-solving error rate on **MATH** test set. The tested models are Llemma-7B (left), Llemma-34B (middle), & Mistral-7B (right). In the legend, M.V. refer to Majority Voting.

Where $Q(s)$ represents the quality score of node $s$, $N(s)$ is the number of visits to node $s$, and $C$ is a constant determining the level of exploration.

**Expansion and evaluation**    Upon reaching a non-terminal node $s$, the node is expanded by generating multiple child nodes. Each child node $c$ is then evaluated using a value function $V(c)$, which predicts the potential quality of continuing the sequence from node $c$.

**Backpropagation**    After evaluation, the algorithm updates the UCT values and the visit counts for all nodes along the path from the selected node back to the root. For any node $n$ in this path, the updates are made as follows:

$$N(n) \leftarrow N(n) + 1,$$
$$Q(n) \leftarrow \frac{(N(n) - 1)\, Q(n) + V(s)}{N(n)}.$$

## C   Hyper-parameters

**Finetuning**    We put all the hyperparameters of fine-tuned models in the table 3. We preprocess the MetaMath Dataset to make the solutions in a stepwise format.

**Inference**    For all the inference strategies, the temperature of the LLM is set to $1.0$. Max tokens for the output is $1024$ and max tokens for one step is $256$. For REBASE, we chose the balance temperature (the softmax temperature in the REBASE algorithm) as $T_b = 0.1$. For MCTS, we set $C$ in the UCT value as 1 and we expand $4, 8, 16$ children for the root, 2 children for other selected nodes with total $32, 64, 128$ expansions respectively.

## D   Supplementary Figures

In the main part of paper, there isn't enough space for showing the scaling effects of Majority Voting, we append the figures about Majority Voting and Majority Voting v.s. Weighted Majority Voting

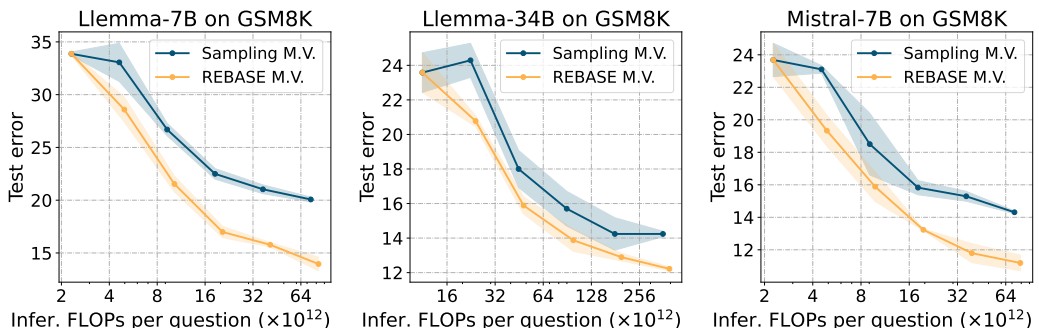

Figure 8: **The inference computation scaling laws** of different models for the problem-solving error rate on **GSM8K** test set. The tested models are Llemma-7B (left), Llemma-34B (middle), & Mistral-7B (right). In the legend, M.V. refer to Majority Voting.

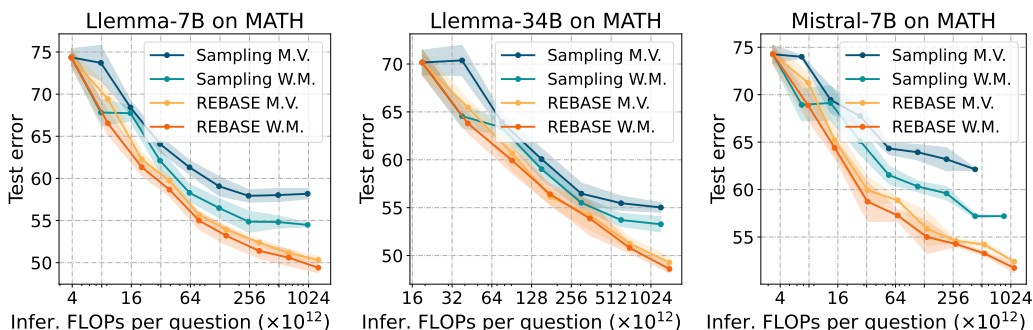

Figure 9: **The inference computation scaling laws** of different models for the problem-solving error rate on **MATH** test set. The tested models are Llemma-7B (left), Llemma-34B (middle), & Mistral-7B (right). In the legend, M.V. and W.M. refer to Majority Voting and Weighted Majority, respectively.

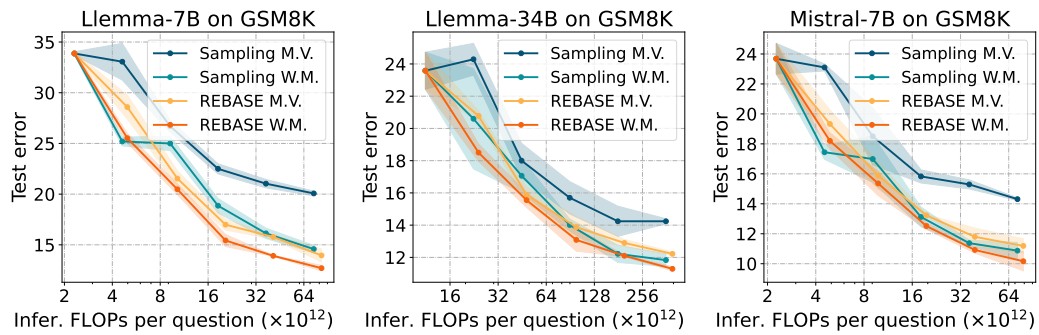

Figure 10: **The inference computation scaling laws** of different models for the problem-solving error rate on **GSM8K** test set. The tested models are Llemma-7B (left), Llemma-34B (middle), & Mistral-7B (right). In the legend, M.V. and W.M. refer to Majority Voting and Weighted Majority, respectively.

(Fig. 7, 8 ,9, 10) in this appendix. The experiments show that although the gap between Majority Voting and Weighted Majority Voting on sampling is huge. This gap becomes much smaller if we apply REBASE. This phenomenon can be caused by the selection ability of tree search like REBASE. Once REBASE already samples solutions with high rewards, conducing weighted majority voting gains less since the sampled solutions may all have relatively high and stable rewards compared with those of sampling.

