# OpenReview forum: "An Empirical Analysis of Compute-Optimal Inference for Problem-Solving with Language Models"
_NeurIPS.cc/2024/Conference — Submitted to NeurIPS 2024_

### Official Review · Reviewer_drh5 · 2024-07-10

**Soundness:** 3
**Presentation:** 3
**Contribution:** 2
**Rating:** 6
**Confidence:** 2

**Summary:**

This paper explores compute-optimal inference for large language
models (LLMs), focusing on designing models and strategies that
balance additional inference-time computation with improved
performance. The study evaluates the effectiveness and efficiency of
various inference strategies, including Greedy Search, Majority
Voting, Best-of-N, and Weighted Voting, across different model sizes
(e.g., 7B and 34B) and computational budgets. Experimental results
indicate that smaller models with advanced tree search algorithms can
achieve a Pareto-optimal trade-off, offering significant benefits for
end-device deployment. For example, the Llemma-7B model matches the
accuracy of the Llemma-34B model on the MATH500 dataset while using
half the FLOPs. These findings suggest that smaller models with
sophisticated decoding algorithms can enhance problem-solving accuracy
across various generation tasks.

**Strengths:**

- The paper focuses on an interesting topic and should be of interest
  to the audience of NeurIPS.
- It considers a comprehensive experimental investigation to confirm
  the claims.
- The proposed tree search algorithm is interesting and seems to
  outperform the competition.

**Weaknesses:**

- Although the paper offers quite thorough experimental analysis, it
  does not look deep in terms of theoretical ideas (although there are
  2 theorems), which may be a problem for a flagship venue like
  NeurIPS.
- Overall findings on the possibility to train an equally accurate
  model with fewer computational resources do not look surprising.
- The paper would benefit from additional proof-reading as there are a
  large number of typos present.

**Questions:**

N/A

**Limitations:**

The paper concentrates on mathematical problem-solving tasks using 7B
and 34B models, with findings potentially not applicable to other
domains. Future research should explore a broader range of model sizes
and different training datasets to better understand compute-optimal
inference in mathematical problem-solving.

I should also say that these limitations have been explicitly
discussed by the authors themselves (so not a criticism).

---

> ### Author Rebuttal · Authors · 2024-08-06
>
> > Concern 1: Although the paper offers quite thorough experimental analysis, it does not look deep in terms of theoretical ideas (although there are 2 theorems), which may be a problem for a flagship venue like NeurIPS.
>
> Our main focus is on formalizing the compute-optimal inference problem, designing a new method, and empirical analysis. To further aid in reasoning about inference strategies in our problem setting, we provided theoretical analysis. We respectfully disagree that these contributions are together insufficient for a venue like NeurIPS. We also follow your suggestion and additionally prove the asymptotic convergence bounds for majority voting and weighted majority voting to understand the performance saturation of the voting methods.
>
> **Notations and assumptions.** Let $\mathcal{V}$ be a _finite_ vocabulary and $\mathcal{V}^*$ its Kleene closure, i.e., the set of all strings. Given a problem $x$, we say a language model answers $y$ to this problem if the model outputs $r\mathrm{e}y$ where $r\in\mathcal{V}^*$ can be any ``reasoning path'' and $\mathrm{e}\in\mathcal{V}$ denotes a special token that marks the end of reasoning. We further assume that the answer string is always shorter than $L$ tokens, i.e., $|y|\leq L$ for some fixed $L\in\mathbb{N}^*$ where $|y|$ denotes the length of $y$.
> For a language model $\pi$, denote by $\pi(v|w)$ the probability of generating $v$ given input (prompt) $w$. For a reward model $\rho$, denote by $\rho(v)$ the score it assigns to the string $v$.
> We use $\mathbb{I}$ to denote the indicator function.
>
> **Theorem.**  Consider a dataset $\mathcal{D}=\\{(x_i, y_i)\\}^m_{i=1}$ where $x_i$ and $y_i$ denote input and true answer, respectively. For a language model $\pi$, denote by $\mathrm{acc}^{\mathrm{MV}}_n (\mathcal{D}; \pi)$ the accuracy on $\mathcal{D}$ using Majority Voting with $n$ samples. Following the notations and assumptions defined above, we have:
>
> $\mathbb{E}\left[\mathrm{acc}_ n^{\mathrm{MV}}(\mathcal{D}; \pi)\right] = \frac{1}{m}\sum_{i=1}^m \mathbb{I}\left[y_i = \arg\max_{|y|\leq L} \sum_{r\in\mathcal{V}^*}\pi(r\mathrm{e}y|x_i)\right] - \mathcal{O}(c^{-n})$  for some constant $c>1$.
>
> For weighted voting, similar convergence results hold, i.e., the accuracy of weighted voting also saturates at the speed of $ \mathcal{O}(c^{-n})$ where $c>1$. We will include these results and proofs in the paper revision.
>
> > Concern 2: Overall findings on the possibility to train an equally accurate model with fewer computational resources do not look surprising
>
> There might be some misunderstanding, since we study the compute-optimal _inference_ strategy in this paper (rather than _training_). Our main findings are: 1. Generally more computation at inference time leads to higher performance, but finally saturates at some point. 2. Using the same training dataset, smaller models can be more compute-optimal than larger models during inference. They typically achieve comparable performance with less computation. 3. While typical tree search algorithms like MCTS are not compute-optimal (in that they may improve performance, but by using much more computation), we found that our proposed new tree search algorithm REBASE can get higher performance with less computation than sampling. This gives the compute-optimal inference strategy: using a smaller model with a sophisticated tree search algorithm (REBASE).
>
>
>
> > Concern 3:The paper would benefit from additional proof-reading as there are a large number of typos present.
>
> Thanks for pointing it out! We have fixed some typos and will do more proof-reading for the final version.
>
>
> We sincerely hope that our responses address your concerns and you reevaluate our work based on the responses. Thank you again for your time!

---

> > ### Comment · Reviewer_drh5 · 2024-08-08
> >
> > Thank you for your response.

---

### Official Review · Reviewer_pgJ7 · 2024-07-12

**Soundness:** 3
**Presentation:** 2
**Contribution:** 2
**Rating:** 5
**Confidence:** 2

**Summary:**

The paper presents an approach to select an optimal inference strategy for LLMs and empirical analysis on Math problem solving tasks. The main idea is to select an inference strategy based on a computational budget (FLOPs). The underlying policy model samples solutions by generating tokens based on the budget and a ranking model consumes these tokens. A new reward model is developed  to explore the solution space more effectively. The reward acts as a weighted majority function over the solutions.
Experiments are performed on Math problem solving benchmarks. Some of the key insights from the experiments is that a smaller LLM can outperform the larger LLM in terms of using a smaller computational budget while maintaining similar accuracy. They also show that the proposed approach with a smaller budget has comparable accuracy than sampling with a larger budget.

**Strengths:**

- The insights that inference time strategy can compensate for using smaller LLMs in generation seems to be interesting
- The experiments also provide a basis for analyzing scaling properties of inference which can be significant

**Weaknesses:**

- In terms of the method itself, I was not sure if it is very novel. It seems to be a smaller variation on the tree search methods that search for solutions in the generated space
- In terms of comparisons, I was not sure about the significance of the benchmark, i.e., are there some properties that make the proposed reward reranking more optimal in Llema model specifically (due to the structure of math problems, etc.). In general, since the main contribution of the paper is empirical, I think there should be experiments or discussions different LLMs to make the contribution more significant.
-Overall, the empirical conclusions seem very tied to the specific benchmarks, so I was a little unsure regarding the significance of the conclusions.

**Questions:**

- Is the comparison based on state of the art inference strategies for compute-optimal inference? Specifically, the other methods are all agnostic of the computational limits, so I was wondering if there are other approaches that do take computational limits into account (the related works do not mention any so it is possible there are not)?

**Limitations:**

Limitations regarding the datasets are mentioned.

---

> ### Author Rebuttal · Authors · 2024-08-07
>
> > Concern 1: In terms of the method itself, I was not sure if it is very novel. It seems to be a smaller variation on the tree search methods that search for solutions in the generated space
>
> Our emphasis in this work is on formulating and studying a new setting of compute-optimal inference. As part of this, we design a new tree search algorithm that performs well in terms of accuracy and compute budget. This results in two kinds of novelty:
>
> - First, we are the first to formulate the compute-optimal inference problem (at least in the context of LLM problem solving). Previous research on inference methods for problem solving mainly targets accuracy improvements and pays little attention to the additional increase in compute budget.
> - Second, we propose the REBASE method which combines the merits of MCTS and sampling in a novel way. REBASE combines a highly parallel tree search with a new branch cut and exploration mechanism. This results in performance improvements from the new exploration mechanism, with a cost comparable to sampling. While on the surface it may appear as a variation of existing tree search methods, it is nontrivial to design such an algorithm and achieve a good accuracy-cost tradeoff. We are not aware of other methods that have done so, and/or used the same mechanisms that we propose.
>
> > Concern 2: In terms of comparisons, I was not sure about the significance of the benchmark, i.e., are there some properties that make the proposed reward reranking more optimal in Llema model specifically (due to the structure of math problems, etc.). In general, since the main contribution of the paper is empirical, I think there should be experiments or discussions different LLMs to make the contribution more significant. -Overall, the empirical conclusions seem very tied to the specific benchmarks, so I was a little unsure regarding the significance of the conclusions.
>
> MATH and GSM8K are the two most widely used benchmarks for evaluating LLM math abilities. Although it is common in recent papers to use only these two benchmarks, to address your concern we additionally provide results on a code generation task (the MBPP benchmark) with different LLMs. Please check the detailed results in the general response.
>
> In our original paper, we discussed various models, including Lemma models and Mistral, which have entirely different architectures. For the new code generation task, in addition to presenting the results from Llama3-8B, we have included the results from CodeLlama-7B-Instruct to demonstrate the effectiveness of REBASE across different LLMs.
>
> **CodeLlama-7B-Instruct**
>
> | Sample num | Sampling FLOPS ($\times 10^{12}$) | Sampling Pass@n | Rebase FLOPS ($\times 10^{12}$) | Rebase Pass@n |
> |------------|------------------------|-----------------|----------------------|---------------|
> | 8          | 13                    | 45.6%           | 10.5                    | 57.6%         |
> | 16         | 26                   | 54.2%           | 21                   |65%         |
> | 32         |52                  | 62.8%             | 42                  | 69%           |
> | 64         | 104                 | 68.6%           | 84                 |72.6%               |
>
>
>
> > Question: Is the comparison based on state of the art inference strategies for compute-optimal inference? Specifically, the other methods are all agnostic of the computational limits, so I was wondering if there are other approaches that do take computational limits into account (the related works do not mention any so it is possible there are not)?
>
> We use state-of-the-art inference strategies (for instance, weighted majority voting and MCTS have been used to achieve state-of-the-art accuracy in problem solving tasks [1-4]), but a key contribution of our paper is pointing out that these methods may not be optimal when cost is taken into account. Since this compute-optimal setting is new when it comes to problem solving, we are not aware of a “state-of-the-art” for compute-optimal inference. Instead, we use strong, commonly used inference strategies, analyze their performance tradeoffs, and show that designing a tree search algorithm with cost in mind can lead to a better tradeoff. The comparison of our Rebase with weighted majority voting and MCTS is shown in figure 1. We also present an example comparison of the Llema 7B performance on the different inference strategies here:
> |inference strategy| FLOPS $\times 10^{13}$|Accuracy on MATH500|
> |----------|--------|-------|
> |weighted majority voting|25.1|45.2%|
> |MCTS|23|44%|
> |Rebase|14.8|46.8%|
>
>
> [1] Wang, Xuezhi, et al. "Self-consistency improves chain of thought reasoning in language models." arXiv preprint arXiv:2203.11171 (2022).
>
> [2] Zhang, Shun, et al. "Planning with large language models for code generation." arXiv preprint arXiv:2303.05510 (2023).
>
> [3] Liu, Jiacheng, et al. "Making ppo even better: Value-guided monte-carlo tree search decoding." arXiv preprint arXiv:2309.15028 (2023).
>
> [4] Tian, Ye, et al. "Toward Self-Improvement of LLMs via Imagination, Searching, and Criticizing." arXiv preprint arXiv:2404.12253 (2024).
>
> We sincerely hope that our responses address your concerns and you reevaluate our work based on the responses. Thank you again for your time!

---

> > ### Comment · Reviewer_pgJ7 · 2024-08-12
> >
> > Thanks for the detailed response. Based on the response, I think the proposed approach seems to yield a state of the art inference strategy. I still think since the main contribution focuses on engineering a better inference strategy and empirical analysis of LLMs, the type of datasets could be broader as suggested by other reviewers as well. At the same time, the proposed approach could improve the usability of LLMs in specific domains (e.g. MATH understanding). I will increase my score based on the discussions.

---

> > > ### Author Response · Authors · 2024-08-12
> > >
> > > Thank you for your response and for taking the time to reevaluate our work. While the rebase method does improve the performances on benchmarks, we would like to clarify that our paper is not primarily focused on proposing new methods. Instead, our key contributions lie in the development of inference scaling laws and the formulation of the inference compute-optimal problem. To illustrate our novel findings, we use the Rebase method, demonstrating that employing smaller models with advanced inference strategies is, in fact, compute-optimal.

---

> > > ### Author Response · Authors · 2024-08-13
> > >
> > > Thank you for increasing our score. Besides the math tasks, we also added benchmark MBPP in code generation, please see the detailed results in our general response.

---

### Official Review · Reviewer_PMgx · 2024-07-13

**Soundness:** 2
**Presentation:** 2
**Contribution:** 2
**Rating:** 4
**Confidence:** 4

**Summary:**

This paper investigates the optimal training configurations of large language models (LLMs) during inference. The proposed inference strategy, REward BAlanced SEarch (REBASE), combines the strengths of Monte Carlo Tree Search (MCTS) with reduced inference costs, resulting in improved performance on math-domain tasks.

**Strengths:**

1. This paper provides a comprehensive overview, i,e, the inference scaling law, of the performance of different sampling strategies under various inference configurations.
2. The novel REBASE inference strategy achieves better downstream task performance under the same computational budget or even less.

**Weaknesses:**

### Major

1. Did you take into account the inference cost of the reward model (RM) in your analysis? As the REBASE frequently uses RM to judge the quality of immediate solutions than other sampling strategies, such as, weighted major voting, It's crucial to consider this aspect to provide a holistic view of the efficiency and practicality of your proposed strategy.

2. The base model with post-training techniques such as SFT and RLHF inherently limits the upper bound of performance. It seems that adding more tricks during inference could improve performance, but the marginal effect may result in diminished returns when using models already tuned by the RLHF process. Could you compare the performance gains of REBASE between the base model, the SFT model, and the Chat model? Is the performance gain only significant in models that have not been tuned?

3. In Section 4.2, the observation in "Scaling law of compute-optimal inference" indicates that the optimal inference strategy is invariant to the amount of compute but depends on the model size, i.e., the model's inherent capacity. This raises a concern: does the inference strategy significantly improve the model's performance, or does it only take effect in certain scenarios, such as with base models that have not been aligned?

4. The paper focuses solely on the math domain. To strengthen your claims, a more comprehensive evaluation across general domains using widely adopted benchmarks, such as MMLU, SuperGLUE, HumanEval, etc,  is necessary.

5. There appears to be no significant improvement in the GSM8K datasets than MATH500 dataset.

### Minor

1. Figures. 2 and 3 are not referenced in the main manuscript.

2. Figures. 2 and 3 appear to be in draft form and are somewhat vague.

**Questions:**

See Weakness.

**Limitations:**

See Weakness.

---

> ### Author Rebuttal · Authors · 2024-08-06
>
> > Concern 1: Did you take into account the inference cost of the reward model (RM) in your analysis? As the REBASE frequently uses RM to judge the quality of immediate solutions than other sampling strategies, such as, weighted major voting, It's crucial to consider this aspect to provide a holistic view of the efficiency and practicality of your proposed strategy.
>
> We didn’t take into account the inference cost of the reward model, because when measuring the inference computation, the cost of the RM is negligible compared to the policy model. While the policy model generates a sequence of tokens in an autoregressive way, the reward model only runs one forward pass to get logits for calculating the score. The dominant part of computation is the decoding process, hence we can only take that into consideration.
>
>
> > Concern 2: The base model with post-training techniques such as SFT and RLHF inherently limits the upper bound of performance. It seems that adding more tricks during inference could improve performance, but the marginal effect may result in diminished returns when using models already tuned by the RLHF process. Could you compare the performance gains of REBASE between the base model, the SFT model, and the Chat model? Is the performance gain only significant in models that have not been tuned?
>
> All of the models in our current paper have been tuned on the GSM8k and MATH training set (i.e., the MetaMath dataset). In order to see how scaling laws of inference vary based on the post-training methodology (or lack thereof),  we conducted additional experiments using Llama3-base and Llama3-Instruct on the MBPP benchmark, with the results shown below.
>
> **Llama-8B-Base**
>
> | Sample num | Sampling FLOPS ($\times 10^{12}$) | Sampling Pass@n | Rebase FLOPS ($\times 10^{12}$) | Rebase Pass@n |
> |------------|------------------------|-----------------|----------------------|---------------|
> | 8          | 5.6                    | 25.8%           | 8                    | 33.2%         |
> | 16         | 11.2                   | 39.8%           | 16                   | 47.4%         |
> | 32         | 22.4                   | 51%             | 32                   | 59%           |
> | 64         | 44.8                   | 62.8%           | 64                   | 68%              |
>
>
> **Llama3-8B-Instruct**
>
> | Sample num | Sampling FLOPS ($10^{12}$) | Sampling Pass@n | Rebase FLOPS ($10^{12}$) | Rebase Pass@n |
> |------------|------------------------|-----------------|----------------------|---------------|
> | 8          | 8                      | 63%             | 8.26                 | 69.6%         |
> | 16         | 16                     | 69.4%           | 17.47                | 72.4%         |
> | 32         | 32                     | 72.4%           | 34.9                 | 75.8%         |
> | 64         | 64                     | 79%             | 69.15                | 81.4%         |
>
>
> We can see that when using REBASE as an inference algorithm, Llama3-8B-Base gets more improvement (7.4%, 7.6%, 8% and 5.2% compared to sampling 8, 16, 32, 64 respectively) than the Llama3-8B-Instruct (6.6%, 3%, 3.4%, 2.4%). This is similar to the “weaker models gain more from the tree search” observation in our paper (section 4.3). However, the performance of Llama3-8B-Instruct–which has undergone several post-training techniques (e.g., SFT, RL)--still significantly improves when using more samples, and the tree-search-based REBASE still outperforms vanilla sampling. Overall thank you for giving this suggestion, which encourages us to conduct extra experiments to strengthen our conclusion that weaker models tend to benefit more from the tree search method.
>
>
> > Concern 3: Section 4.2, the observation in "Scaling law of compute-optimal inference" indicates that the optimal inference strategy is invariant to the amount of compute but depends on the model size, i.e., the model's inherent capacity. This raises a concern: does the inference strategy significantly improve the model's performance, or does it only take effect in certain scenarios, such as with base models that have not been aligned?
>
>
> In the paper, all the models (Llemma and mistral) are SFT-aligned. To further address your concern, we provide new experiment results showing that the inference strategies also take effect in RL-tuned models like Llama3-Instruct-8B.
>
> > Concern 4: The paper focuses solely on the math domain. To strengthen your claims, a more comprehensive evaluation across general domains using widely adopted benchmarks, such as MMLU, SuperGLUE, HumanEval, etc, is necessary.
>
> We conduct additional experiments on MBPP to show our findings are also applicable to code generation.  Rebase consistently outperforms the sampling method, the accuracy raises from 62.8% to 68% and 79% to 81.4% for Llama3-base and Llama3-Instruct when sampling 64 solutions.
>
>
> > Concern 5: There appears to be no significant improvement in the GSM8K datasets than MATH500 dataset.
>
> This is because GSM8K is much easier than MATH500, so the error rate ranges of these two datasets are different. In Table 2, we can see that GSM8K’s error rate is typically around 10-15%, so 0.7-1.9% are already relatively significant improvements. For MATH500, the error rate is around 55-60%, so that we can achieve an absolute accuracy improvement of 2.6-5.3%.
>
>
> We sincerely hope that our responses address your concerns and you reevaluate our work based on the responses. Thank you again for your time!

---

> > ### Comment · Reviewer_PMgx · 2024-08-09
> >
> > Thank you for your detailed responses. However, I'm still uncertain about the practical implications of the inference scaling law. It seems more appropriate to apply the RM during the post-training stage rather than the inference stage, as it doesn't enhance the model's inherent capacity. Additionally, the performance improvements don't seem significantly better than the vanilla inference strategy. Thus, I  choose to retain my score.

---

> > > ### Author Response · Authors · 2024-08-09
> > >
> > > Thank you for the feedback. We respond to your follow-up concerns below.
> > >
> > > > uncertain about the practical implications of the inference scaling law
> > >
> > > Existing research shows that either scaling up model sizes or applying advanced inference strategies improves LLMs’ task performance. However, given an inference compute budget, it is not clear whether using large models with simple inference strategies or small models with sophisticated inference strategies is more favorable. We believe that identifying the right model size and inference strategy is as important as, if not more important than, the research on training scaling law [1] which studies the trade-off between model sizes and training number of tokens.
> > >
> > > Our empirical analysis reveals that using smaller models with advanced inference strategy (like Rebase) is compute-optimal under budget constraints. We note that this is a novel and useful finding which provides important guidance to model deployment on end devices.
> > >
> > > [1] Hoffmann, Jordan, et al. "Training compute-optimal large language models."
> > >
> > > > It seems more appropriate to apply the RM during the post-training stage rather than the inference stage
> > >
> > > We feel that there may be some misunderstandings regarding the RMs. The RM in the RLHF scores the whole LLM output by learning from human feedback. In contrast, Rebase uses the process reward model (PRM) [2-3] which scores each reasoning step and navigates for a better path of reasoning. Thus, **both the training methods and working mechanisms of the two “RM”s in Rebase and RLHF are different.** We will clarify this in our paper.
> > >
> > > What’s more, recent research has shown that the RL-aligned (with RM) model with vanilla inference strategy underperforms the model using inference strategies with RMs. [4] shows that RL-aligned Llemma 7b mode with the reward model has 34.0% accuracy (greedy decoding), which underperforms the weight majority voting of corresponding non-RL model(42.7%). [5] shows that after alignment against a reward model, the policy model still benefits from reward-model-based tree search, with the performance improved from 61.84% to 67.04% on MATH dataset. Thus, we believe that **using RM in the post-training / inference stage can be two orthogonal choices** to improve performance.
> > >
> > > [2] Lightman, Hunter, et al. "Let's verify step by step."
> > >
> > > [3] Ma, Qianli, et al. "Let's reward step by step: Step-Level reward model as the Navigators for Reasoning."
> > >
> > > [4] Sun, Zhiqing, et al. "Easy-to-hard generalization: Scalable alignment beyond human supervision."
> > >
> > > [5] Chen, Guoxin, et al. "AlphaMath Almost Zero: process Supervision without process."
> > >
> > > > The Inference strategy doesn’t enhance the model’s inherent capability.
> > >
> > > Besides enhancing the model’s inherent capability, inference-time improvements are also important since our ultimate goal is to achieve better task performances. Generally learning and inference are complementary methods [6] for improving LLMs task performances. There are many works focused on inference improvement. For example, [7] introduces search during inference, [8] refine the LLMs output to get better results and [9] integrates the LLM’s output with programs to enhance the reasoning capability.
> > >
> > > [6] Jones, Andy L. "Scaling scaling laws with board games."
> > >
> > > [7] Yao, Shunyu, et al. "Tree of thoughts: Deliberate problem solving with large language models." NeurIPS 2024.
> > >
> > > [8]  Madaan, Aman, et al. "Self-refine: Iterative refinement with self-feedback." NeurIPS 2024.
> > >
> > > [9] Gao, Luyu, et al. "Pal: Program-aided language models." International Conference on Machine Learning. PMLR, 2023.
> > >
> > > > The performance improvements don’t seem significantly better than the vanilla inference strategy.
> > >
> > > In our general response, one can see that Rebase leads to huge improvements on the accuracy rate. Specifically, Rebase improves the baseline performance by 5.2%-8% (Llama3-Base) and 2.4%-6.6%(Llama3-Instruct) under different sampling numbers.
> > >
> > > In our paper, we compare the Rebase with weighted majority voting in the plots and paper, where both rebase and weighted majority voting have used the reward model. To compare the performance of Rebase and vallina inference strategy (majority voting or greedy decoding), we present the results here:
> > >
> > > **Mistral 7B on MATH**
> > >
> > > |   |greedy decoding|Maj@256|Rebase@256|
> > > |---|-------|------|------|
> > > |Accuracy|28.6%  |  37.8% |  46.8% |
> > >
> > > **Llemma 7b on MATH dataset**
> > > |   |greedy decoding|Maj@256|Rebase@256|
> > > |---|-------|------|------|
> > > |Accuracy|30%  |  41.8% |  50.6% |
> > >
> > > **Llemma 34b on MATH dataset**
> > > |   |greedy decoding|Maj@64|Rebase@64|
> > > |---|------|-------|-------|
> > > |Accuracy|33%  |  45% |  51.4% |
> > >
> > > Rebase improves over greedy decoding by ~20 points, and naive majority voting by 8 points on average. We believe those performance gains are significant.
> > >
> > > We hope our responses address your concerns and you reconsider the score. Thank you for your time!

---

> > > ### Author Response · Authors · 2024-08-13
> > >
> > > Thank you for your review. We would appreciate it if you could confirm whether our response has addressed your additional concerns. We have clarified the practical implications of inference scaling, the distinction between PRM in inference and RM in RLHF, and the significance of inference-time algorithms in the last thread. Please let us know if you have any further questions, we hope you might consider increasing the score, thanks!

---

### Author Rebuttal · Authors · 2024-08-06

**General Response**

We are grateful to all reviews for their insightful comments. We appreciate that reviewers found our method to be novel (PMgX), basis for analyzing inference scaling law to be comprehensive (PMgX, pgJ7, drh5),  and our topic to be interesting (pgJ7, drh5).

We summarize our contributions and add new experiments as your suggestions here:

**Key contributions**:
-  We are the first to formulate the compute-optimal inference problem and conduct a comprehensive study of different model sizes and inference strategies.
- Through our comprehensive study on the compute-optimal inference problem, we find that increasing computational budget during inference generally leads to enhanced performance. Additionally, we demonstrate an upper bound for the voting method and show that advanced inference strategies (weighted voting, Rebase) offer a better trade-off between computation and performance.
- While previous tree search methods sacrifice computation cost for performance improvement, our proposed Rebase method is a compute optimal inference strategy which achieves high performance with less computation.
- We find that using smaller models with sophisticated inference strategies is the compute-optimal approach for inference.


**New Experiments**:

Following the suggestions of studying other benchmarks (PMgX, pgJ7) and comparing the performance of the base model with the rl-tuned model (PMgX), we add new experiments on the code-generation benchmark MBPP with Llama3-base, Llama3-Instruct models. For rebase and sampling, we use the same configuration (prompt (zero shot), temperature, top_p, etc).

**Llama-8B-Base**

| Sample num | Sampling FLOPS ($\times 10^{12}$) | Sampling Pass@n | Rebase FLOPS ($\times 10^{12}$) | Rebase Pass@n |
|------------|------------------------|-----------------|----------------------|---------------|
| 8          | 5.6                    | 25.8%           | 8                    | 33.2%         |
| 16         | 11.2                   | 39.8%           | 16                   | 47.4%         |
| 32         | 22.4                   | 51%             | 32                   | 59%           |
| 64         | 44.8                   | 62.8%           | 64                   | 68%              |




**Llama3-8B-Instruct**

| Sample num | Sampling FLOPS ($\times 10^{12}$) | Sampling Pass@n | Rebase FLOPS ($\times 10^{12}$) | Rebase Pass@n |
|------------|------------------------|-----------------|----------------------|---------------|
| 8          | 8                      | 63%             | 8.26                 | 69.6%         |
| 16         | 16                     | 69.4%           | 17.47                | 72.4%         |
| 32         | 32                     | 72.4%           | 34.9                 | 75.8%         |
| 64         | 64                     | 79%             | 69.15                | 81.4%         |


These new results strengthen our claim that Rebase is a compute-optimal inference approach. Namely, Rebase consistently outperforms the baseline across the different models (base, sft, rl). Also, the new results further clarify the finding that weaker models gain more and stronger models gain less (this is also pointed out in section 4.3 of our paper).

We also find that when the computation is limited and a naive sampling method is used, the gap between the base model and rl-instruct model is huge (25.8% v.s. 63%), but with more computation and an advanced inference strategy, the gap is narrowed (68% v.s. 81.4%). This suggests that the inference-time strategy can bridge the gap between strong and weak models.

---

### Decision · Program_Chairs · 2024-09-25

**Decision:**

Reject

**Comment:**

The paper explores how best to balance LLM model size and the subsequent inference (by RL) to improve accuracy. The method “Reward Balanced Search (REBASE) is proposed In the context of MCTS balances the expansion width among nodes at the same depth based on the rewards given by Process reward model (PRM).

Reviewers were overall  mixed but the majority tended to reject.. The discussion did convinced one reviewer to raise its rank but another reviewer remained highly critical. There was no further discussion beyond the discussion with the authors. The issues raised had to do with how novel is the approach and whether or not the empirical results are impressive enough, if there is comparison to state of the art solvers, or earlier solvers and the scope of the work (focused on Math problems).  Reading the paper myself I think it lacked a clear problem statement and context of what is the underlying problem that is attempted to be solved here.  Is the context solving RL instances? MDPsd? In that sense the paper was not self-contained which made it hard to evaluate.